# VERSATILE NEURAL PROCESSES FOR LEARNING IMPLICIT NEURAL REPRESENTATIONS

**Zongyu Guo**[1][*], **Cuiling Lan**[2] , **Zhizheng Zhang**[2] , **Yan Lu**[2] , **Zhibo Chen**[1]
[1]University of Science and Technology of China, [2]Microsoft Research Asia
[1] `{guozy@mail., chenzhibo@}ustc.edu.cn`
[2] `{culan, zhizzhang, yanlu}@microsoft.com`

## ABSTRACT

Representing a signal as a continuous function parameterized by neural network (a.k.a. Implicit Neural Representations, INRs) has attracted increasing attention in recent years. Neural Processes (NPs), which model the distributions over functions conditioned on partial observations (context set), provide a practical solution for fast inference of continuous functions. However, existing NP architectures suffer from inferior modeling capability for complex signals. In this paper, we propose an efficient NP framework dubbed Versatile Neural Processes (VNP), which largely increases the capability of approximating functions. Specifically, we introduce a bottleneck encoder that produces fewer and informative context tokens, relieving the high computational cost while providing high modeling capability. At the decoder side, we hierarchically learn multiple global latent variables that jointly model the global structure and the uncertainty of a function, enabling our model to capture the distribution of complex signals. We demonstrate the effectiveness of the proposed VNP on a variety of tasks involving 1D, 2D and 3D signals. Particularly, our method shows promise in learning accurate INRs w.r.t. a 3D scene without further finetuning. Code is available here.

## 1 INTRODUCTION

A recent line of research on learning representations is to model a signal (*e.g.*, image, 3D scene) as a continuous function that map the input coordinates into the corresponding signal values. By parameterizing a continuous function with neural networks, such implicitly defined representations, *i.e.*, implicit neural representations (INRs), offer many benefits over conventional discrete (*e.g.*, grid-based) representations, such as the compactness and memory-efficiency (Sitzmann et al., 2020b; Tancik et al., 2020; Mildenhall et al., 2020; Chen et al., 2021a). Characterizing/parameterizing a signal by a corresponding set of network parameters generally requires re-training the neural network, which is computationally costly. In practice, at test time, it is desired to have models that support fast adaptation to partial observations of a new signal without finetuning.

In fact, the Neural Processes (NPs) family (Jha et al., 2022) supports such merit. It meta-learns the implicit neural representations of a probabilistic function conditioned on partial signal observations. During test-time inference, it enables the prediction of the function values at target points within a single forward pass. Naturally, given partial observations of a signal, there exists uncertainty inside its continuous function since there are many possible ways to interpret these observations (*i.e.*, context set). The NP methods (Garnelo et al., 2018a;b) learn to map a context set of observed input-output pairs to a conditional distribution over functions (with uncertainty modeling). However, it has been observed that NPs are prone to underfit the data distribution. Following the spirits of variational auto-encoders (Kingma & Welling, 2014), the work of (Garnelo et al., 2018b) introduces a global latent variable to better capture the uncertainty in the overall structure of the function, which still suffers from the inferior capability for modeling complex signals. Attentive Neural Processes (ANP) (Kim et al., 2019) can further alleviate this issue, which leverages the permutation-invariant attention mechanism (Vaswani et al., 2017) to reweight the context points and the target predictions. However, taking each context point as a token, ANP has troubles in processing complex signals that requires

---
[*]Work done during an internship at Microsoft Research Asia.

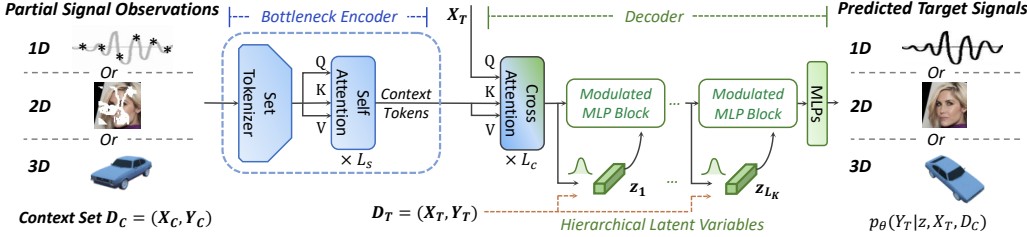

Figure 1: The proposed Versatile Neural Processes framework contains a bottleneck encoder and a hierarchical latent modulated decoder. The input context set is first encoded into fewer and informative context tokens by a set tokenizer followed by self-attention blocks, which provide powerful network capability with tolerable complexity. The decoder consists of cross-attention modules and multiple modulated MLP blocks, enhancing the model expressiveness for complex signals.

abundant context points as condition (*e.g.*, image with high resolution), where the computational cost is very expensive. Moreover, for complex signals, modeling the global structure and uncertainty of the function with a single latent Gaussian variable may be suboptimal. It is worthwhile to explore an efficient framework to excavate the potential of NPs in modeling complex signals.

In this paper, we propose Versatile Neural Processes (VNP), an efficient and flexible framework for meta-learning of implicit neural representations. Figure 1 shows the framework of VNP. Specifically, VNP consists of a bottleneck encoder and a hierarchical latent modulated decoder. The bottleneck encoder powered by the set tokenizer and self-attention blocks encodes the set of context points into fewer and informative context tokens, refraining from high computational cost especially on complex signals while attaining higher modeling capability. At the decoder, we hierarchically learn multiple latent Gaussian variables for jointly modeling the global structure and uncertainty of the function distributions. Particularly, we sample from the latent variables and use them to modulate the parameters of the MLP modules. Our VNP has high expressiveness to complex signals (*e.g.*, 2D images and 3D scenes) and significantly outperforms existing NPs approaches on 1D synthetic data.

We summarize our main contributions as below:

- We propose Versatile Neural Processes (VNP) that is capable of learning accurate INRs for approximating the function of a complex signal.
- We introduce a bottleneck encoder to produce compact yet representative context tokens, facilitating the processing of complex signals with tolerable computational complexity.
- We design a hierarchical latent modulated decoder that can better capture and describe the structure and uncertainty of functions through the joint modulation from the multiple global latent variables.
- We implement the VNP framework on 1D, 2D, and 3D signals respectively, demonstrating the state-of-the-art performance on a variety of tasks. Particularly, our method shows promise in learning accurate INRs of 3D scenes without further finetuning.

## 2 RELATED WORK

**Implicit Neural Representations (INRs).** INRs aim at parameterizing a signal by a differentiable neural network, *i.e.*, learning a *continuous* mapping function *w.r.t.* the signal (Stanley, 2007; Sitzmann et al., 2020b; Tancik et al., 2020). In the seminal work CPPN (Stanley, 2007), a neural network is trained to learn the implicit function that fits a signal, *e.g.*, an image. Given any spatial position identified by a 2D coordinate, the model that acts as a function, outputs the color value of this position. Such continuous representations, as a powerful paradigm, have a wide range of applications such as image super-resolution (Chen et al., 2021b), modeling shapes (Chen & Zhang, 2019; Park et al., 2019) and textures (Oechsle et al., 2019), 3D scene reconstruction (Mildenhall et al., 2020; Martin-Brualla et al., 2021; Niemeyer & Geiger, 2021), and even lossy compression (Dupont et al., 2021; 2022; Schwarz & Teh, 2022). Most of these methods require re-training the neural network to model/overfit

a new signal, which is computationally costly (Sitzmann et al., 2020b; Tancik et al., 2020; Mildenhall et al., 2020; Chen et al., 2021a; Dupont et al., 2021).

In practice, it is desired to have models that support fast adaptation to a new signal, *i.e.*, approaching the continuous function of this signal without abundant steps of optimization in inference. Some works (Chen et al., 2021b; Lee et al., 2021; Sitzmann et al., 2020a) adopt standard gradient-based meta-learning algorithms to learn the initial weight parameters of the network (Finn et al., 2017). However, they still require a few gradient decent steps to fit for the new signals.

**Neural Processes (NPs).** Neural Processes actually can learn the continuous function conditioned on partial observations of a signal, enabling fast adaptation to a new signal (not requiring finetuning in inference). The series of NPs methods (Garnelo et al., 2018a;b) provide probabilistic solutions in predicting continuous functions from partial observations. NPs approximate the distributions in the function space, which formulates Stochastic Processes (Ross et al., 1996), by introducing stochasticity in function realization. A line of researches on neural processes has been introduced, targeting at improving the prediction accuracy (Kim et al., 2019; Lee et al., 2020; Wang & Van Hoof, 2020; Volpp et al., 2021), preserving the stationarity of stochastic processes (Gordon et al.; Foong et al., 2020), and generalizing to observation noise (Kim et al., 2022). The work of Neural Processes (Garnelo et al., 2018b) learns a latent variable distribution to capture the global uncertainty in the overall structure of the function, which is optimized with variational inferece (Kingma & Welling, 2014). Attentive Neural Processes (ANP) leverages the attention mechanism to enhance the representation of each context point and alleviate the underfitting problem (Kim et al., 2019). Transformer Neural Processes (TNP) (Nguyen & Grover, 2022) similarly takes each context point as a token and leverages transformer architecture to approximate the function. However, for complex signals that requiring abundant context points as condition (*e.g.*, image with high resolution), the computational complexities of ANP and TNP are very high which are quadratic with respect to the number of context points.

It is desired to have a framework that can effectively approximate the functions of complex signals. In this work, we introduce a strong NP framework, Versatile Neural Processes (VNP), which leverages informative context tokens and explores the hierarchical global latent variables for modulation, leading to superior approximation of function distributions.

## 3 REVISITING THE PROBLEM FORMULATION OF NPS

Neural Processes (NPs) (Garnelo et al., 2018b;a) are a class of methods that approximate the probabilistic distribution of continuous functions conditioned on partial observations. Suppose we have a labeled context set $D_C = (X_C, Y_C) := (\mathbf{x}_i, \mathbf{y}_i)_{i \in C}$ sampled from a continuous function with inputs $\mathbf{x}_i$ and outputs $\mathbf{y}_i$. NPs target at predicting arbitrary and finite target points $D_T = (X_T, Y_T) := (\mathbf{x}_i, \mathbf{y}_i)_{i \in T}$ by learning the input-output mapping function $f$. Given some signal observations (a set of context points), many possible functions may match well to these observations and thus naturally there exists function uncertainties. The conditional distributions of targets points can be modeled as:

$$p_\phi(Y_T|X_T, D_c) = \prod_{(\mathbf{x},\mathbf{y}) \in D_T} \mathcal{N}(\mathbf{y}; \mu_\mathbf{y}(\mathbf{x}, D_c), \sigma_\mathbf{y}^2(\mathbf{x}, D_c)). \tag{1}$$

To generate coherent function predictions and better model the function distributions, Garnelo *et al.* (Garnelo et al., 2018b) introduce the (Latent) Neural Process by encoding the global structure and uncertainty of the function into a latent Gaussian variable via Bayesian inference:

$$\mathbf{z} \sim p_\theta(\mathbf{z}|X_T, D_C); \quad p_{\phi,\theta}(Y_T|X_T, \mathbf{z}) = \prod_{(\mathbf{x},\mathbf{y}) \in D_T} \mathcal{N}(\mathbf{y}; \mu_\mathbf{y}(\mathbf{z}, X_T, D_C), \sigma_\mathbf{y}^2(\mathbf{z}, X_T, D_C)). \tag{2}$$

Due to the intractable log-likelihood, some previous works adopt amortized variational inference (Kingma & Welling, 2014), which is also used to optimize our proposed framework. We can derive the evidence lower bound (ELBO) on $\log p_\theta(Y_T|X_T, \mathbf{z})$, where the ELBO can be viewed as a combination of the reconstruction term (the first term) and the KL term (the second term):

$$\mathbb{E}_{\mathbf{z} \sim q_\phi(\mathbf{z}|D_T)}[\log p_\theta(Y_T|\mathbf{z}, X_T, D_C)] - D_{KL}[q_\phi(\mathbf{z}|D_T)||p_\psi(\mathbf{z}|X_T, D_C)]. \tag{3}$$

Here, $\psi$ and $\theta$ refer to the parameters of conditional prior encoder and the decoder, similar to conditional VAEs (Sohn et al., 2015; Ivanov et al., 2019). $q_\phi(\mathbf{z}|D_T)$ denotes the posterior distribution of the latent variable given the ground truth target points, which is only used in training and not accessed in inference.

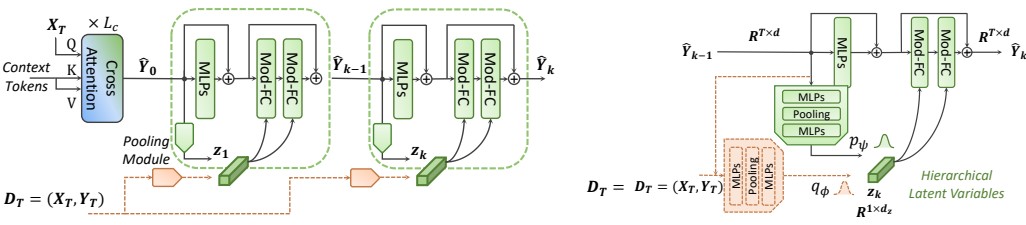

(a) Pipeline of the decoder.  (b) Modulated MLP block.

Figure 2: Diagram of the decoder with the hierarchical latent modulated MLPs. .

So far, we have revisited the problem formulation of NPs. Since NPs approximate the distribution over functions with some function observations, they can also be regarded as *probabilistic, continuous, conditional* generative models, where the generated result is a continuous function.

## 4 VERSATILE NEURAL PROCESSES

We propose an efficient NP framework, Versatile Neural Processes (VNP), which provides much improved capability in approximating the function of a various signal. Figure 1 provides an overview of our proposed VNP. VNP is generic that can be used to generate the functions for 1D, 2D or 3D signals. For test-time inference, the inputs are a set of context points $D_C = (X_C, Y_C)$ and the coordinates of target points $X_T$. The outputs are the predicted values of target points $Y_T$. The continuous function is parameterized by the network weights. The proposed VNP consists of a bottleneck encoder that efficiently encodes the context points into fewer informative context tokens, refraining from high computational cost especially on complex signals. At the decoder side, cross-attention layers are utilized to exploit the contexts relevant to the given target followed by a stack of modulated MLP blocks. Particularly, we introduce multiple global latent variables to jointly modulate the MLP parameters, which facilitates the modeling of complex signal.

### 4.1 BOTTLENECK ENCODER IN VNP

Some NP methods encode the context points to produce local feature representations at target coordinates. The early work (Garnelo et al., 2018b) uses simple MLP layers to learn features of the context set and suffers from underfitting problem. Attentive NP (Kim et al., 2019) and Transformer NP (Nguyen & Grover, 2022) enhances the feature representation by introducing point-wise self-attention with each context point taken as a token. However, when the number of context points is large (*e.g.* in order to model an image signal with many details), the computational burden is heavy and such a framework is impractical.

We address this issue by simply introducing a set tokenization module (set tokenizer) followed by self-attention layers. The set tokenizer is instantiated by a set convolution layer (Zaheer et al., 2017) which transforms the neighboring context points to a token. Taking a 2D image as an example, with the kernel size of $k \times k$ and stride of $k$, the set tokenizer can reduce the number of sample points by $k^2$ times (assume the image resolution is an integer multiple of $k$). By using the set tokenizer, the complexity of attention layers is reduced from the first term to the second term as below:

$$\mathcal{O}(L_s N_C^2 + L_c N_C N_T) \rightarrow \mathcal{O}(L_s \frac{N_C^2}{k^4} + L_c \frac{N_C}{k^2} N_T), \tag{4}$$

where $L_s$ and $L_c$ are the number of the self-attention layers and the cross-attention layers, $N_C$ and $N_T$ are the number of context points and target points. The difference between the set convolution and the conventional convolution is that the former can handle missing points (*e.g.*, in the application of inpainting) and the data that live "off the grid" (*e.g.* time series data that observed irregularly at any time). The combination of set tokenizer and attention would empower our framework the flexibility in processing 3D signals, which is infeasible for ConvNP (Gordon et al.; Foong et al., 2020) because ConvNP requires to preserve all the 3D grids, which is very inefficient in sparse 3D space.

## 4.2 Hierarchical Latent Modulated MLPs

In the previous NP methods (Garnelo et al., 2018b; Kim et al., 2019), they learn a *single* global latent Gaussian variable from the observed context set to model the distributions of a function. However, it potentially limits the expressiveness of the model. Intuitively, increasing the dimension of the latent variables may increase such flexibility, but in practice, this is not sufficient (Lee et al., 2020).

We design a hierarchical latent modulated decoder in order to better model the complex distribution of a function. We sequentially learn multiple global latent variables to modulate the parameters of MLP blocks. Figure 2 shows the details of the designed hierarchical structure.

The decoder consists of $L_c$ cross-attention blocks and $L_K$ modulated MLP blocks. With the target location $X_T$ as query and the context tokens from encoder as the keys and values, the cross-attention blocks output the target location features $\hat{Y}_0 \in \mathbb{R}^{T \times d}$, where $T$ is the number of target coordinates and $d$ is the feature dimension. The sequential modulated MLP blocks enable the exploitation of hierarchical global latent variables for the approximation of a complex signal. The output of the final ($L_K^{th}$) modulated MLP block goes through two MLP layers to estimate the probability of $Y_T$.

Figure 2b illustrates the detailed network structure of a modulated MLP block. We build a Modulated MLP block by stacking two modulated MLP layers and two unmodulated MLP layers. This block refines $\hat{Y}_{k-1}$ to output features $\hat{Y}_k$, which is the input of the next Modulated MLP block. At the heart of each block is a latent variable $\mathbf{z} \in \mathbb{R}^{1 \times d_{\mathbf{z}}}$ modeled by a Gaussian distribution. The generation process of $\mathbf{z}_k$ (of the $k^{th}$ block) can be formulated as follows:

$$p_\psi(\mathbf{z}_k|\hat{Y}_{<k}, D_C, X_T) = \mathcal{N}(\mu_{\mathbf{z_k}}, \sigma_{\mathbf{z_k}}^2) \leftarrow \text{MLPs} \left( \text{AvgPool} \left( \text{MLPs}( \hat{Y}_{k-1} ) \right) \right), \quad (5)$$

where MLPs refer to two MLP layers and an intermediate ReLU activation layer. The prediction results $\hat{Y}_{k-1}$ from the previous block (*i.e.*, the $(k-1)^{th}$ block) are used for calculating the conditional prior distribution of $\mathbf{z}_k$, *i.e.*, $p_\psi(\mathbf{z}_k|\hat{Y}_{<k}, D_C, X_T)$. During training, the conditional posterior distribution $q_\phi(\mathbf{z}_k|\hat{Y}_{<k}, D_C, D_T)$ can be calculated as well, by incorporating the ground truth target signal values $Y_T$:

$$q_\phi(\mathbf{z}_k|\hat{Y}_{<k}, D_C, D_T) = \mathcal{N}(\mu_{\mathbf{z_k}}, \sigma_{\mathbf{z_k}}^2) \leftarrow \text{MLPs} \left( \text{AvgPool} \left( \text{MLPs}([\hat{Y}_{k-1}, D_T]) \right) \right). \quad (6)$$

Marked with dashed lines, $Y_T$ only participates in training and cannot be accessed during inference.

After sampling the low-dimensional latent variable $\mathbf{z}_k$, we use the modulated fully-connected (ModFC) layer (Karras et al., 2020; 2021) to adjust the parameters of modulated MLP layers, taking a sampled realization of $\mathbf{z_k}$ as the style vector input. Unlike previous NP methods that concatenate the latent variable to every target coordinate, modulating the MLP parameters from the low-dimensional latent variables is a more flexible way to represent continuous functions (Sitzmann et al., 2020b). Please see Appendix A for more details for the mechanism of the modulated MLP.

We model the function representations by multiple latent variables $(\mathbf{z}_1, \mathbf{z}_2, ..., \mathbf{z}_{L_K})$. The KL term in Eq. 3, which measures the mismatch between the approximated distribution and ground truth distribution, takes the hierarchical format as

$$D_{KL} = \sum_{k=2}^{L_K} \mathbb{E}_{q_\phi(\mathbf{z}_{<k}|D_C, D_T)}[D_{KL}[q_\phi(\mathbf{z}_k|\hat{Y}_{<k}, D_C, D_T)||p_\psi(\mathbf{z}_k|\hat{Y}_{<k}, D_C, X_T)]]$$
$$+ D_{KL}[q_\phi(\mathbf{z}_1|D_C, D_T)||p_\psi(\mathbf{z}_1|D_C, X_T)], \quad (7)$$

where $q_\phi(\mathbf{z}_{<k}|D_C, D_T) = \prod_{i=1}^{k-1} q_\phi(\mathbf{z}_i|\hat{Y}_{<i}, D_C, D_T)$ is the approximate posterior of $\mathbf{z}_{<k}$. This decomposed KL term and the reconstruction term formulate the objective:

$$\mathcal{L} = \mathbb{E}_{\mathbf{z}_{1:L_K} \sim q_\phi(\mathbf{z}_{1:L_K}|D_T)}[-\log p_\theta(Y_T|\mathbf{z}_{1:L_K}, X_T, D_C)] + \beta \cdot D_{KL}, \quad (8)$$

where $\beta$ denotes a weight to balance the importance between the two terms. In our experiments on 2D and 3D signals, we multiply the KL term with a small weight $\beta$ to better capture the uncertainty of function distributions (Higgins et al.). In addition, we emphasize that although there are some prior works designing hierarchical architecture for VAEs (Sønderby et al., 2016; Vahdat & Kautz, 2020; Child, 2021) or double latent variable models for NP (Wang & Van Hoof, 2020), our proposed hierarchical architecture is different in principle. Here, every latent variable is a low-dimensional vector obtained after average pooling. Therefore, the designed hierarchical architecture can deal with arbitrary target coordinates and thereby can be used for boosting the performance of approximating the global structure of continuous functions.

| | RBF kernel GP | | Matern kernel GP | | Parameters |
|---|---|---|---|---|---|
| | context | target | context | target | |
| CNP | 1.023±0.033 | 0.019±0.015 | 0.935±0.036 | -0.124±0.010 | 0.99 M |
| BANP | 1.380±0.000 | 0.267±0.001 | 1.380±0.002 | 0.072±0.002 | 1.58 M |
| ConvNP | 1.382±0.001 | 0.275±0.001 | 1.383±0.001 | 0.081±0.008 | 1.97 M |
| Stacked ANP | 1.381±0.001 | 0.400±0.004 | 1.381±0.001 | 0.183±0.012 | 1.52 M |
| Stacked ANP + | 1.381±0.001 | 0.406±0.006 | 1.381±0.001 | 0.188±0.014 | 2.31 M |
| HNP | 1.374±0.002 | 0.561±0.003 | 1.377±0.001 | 0.336±0.008 | 1.66 M |
| HNP-Mod | 1.379±0.001 | 0.627±0.020 | 1.371±0.004 | 0.370±0.023 | 1.96 M |
| VNP | 1.377±0.004 | **0.651±0.001** | 1.376±0.004 | **0.439±0.007** | 2.29 M |

Table 1: The test log likelihood (larger is better) on the synthetic 1D regression experiment. The proposed VNP outperforms previous methods by a large margin. Both the pre-processing transformer and the hierarchical structure improve the expressiveness of function representations. Stacked ANP + means the enhanced version of Stacked ANP with more channels.

## 5 EXPERIMENTS

The proposed Versatile Neural Process (VNP), as an efficient meta-learner of implicit neural representations, can be implemented into a variety of tasks. We evaluate the effectiveness of VNP on 1D function regression (subsection 5.1), 2D image completion and superresolution (subsection 5.2), and view synthesis for 3D scenes (subsection 5.3), respectively.

### 5.1 1D SIGNAL REGRESSION

We implement the proposed VNP to learn the implicit neural representations for 1D signal regression. This classical 1D regression aims at predicting the function values at given target locations, conditioned on several observations of the samples from the function. Following (Kim et al., 2019), we measure the performance by considering the context set likelihood and target set likelihood, which reflects the context reconstruction error and target prediction error, respectively.

**Settings.** We train the models on synthetic functions drawn from prior function distributions synthesized with different kernels (RBF, Matern) by following (Gordon et al.; Kim et al., 2022). For the evaluated methods, we employ importance weighted sampling (Burda et al., 2016) to evaluate the log likelihood, where the last four methods in Table 1 are measured by sampling latent variables from the posterior distribution and calculating the tighter ELBO with importance weighted sampling. For fair comparison, we manage to keep the network size comparable with that of other methods, which is completed by adjusting channel number. Please refer to Appendix B for detailed settings.

**Comparison with the State-of-the-Arts**. We compare our method with conditional neural process (CNP) (Garnelo et al., 2018a), (stacked) attentive neural process (ANP) (Kim et al., 2019) (with stacked self-attention layers), bootstrapping attentive neural process (BANP) (Lee et al., 2020), and convolutional neural process (ConvNP) (Foong et al., 2020). Table 1 shows the results in terms of log likelihood. Most of the models can provide satisfactory reconstruction results for context points, except CNP which suffers from underfitting problem (Garnelo et al., 2018b). On the context points, ours also achieves comparable performance. However, the prediction results of other methods on the unseen target points are much worse than that on the context sets. In contrast, our final model, VNP, outperforms previous approaches by a large margin at the target points on both of the function distributions. Note that although BANP attempts to increase the expressiveness of function representations by using latent variable bootstrapping from the perspective data resampling.

**Ablation Study.** We conduct a group of ablation study on this 1D regression task to investigate the effectiveness of different components. Based on ANP, we first build a Hierarchical Neural Process (HNP) with hierarchical latent variables. Note that similar to ANP, the implemented HNP still concatenates every target coordinate with the global latent variables. We observe significant improvements in Table 1 by comparing HNP with ANP, demonstrating that the hierarchical global latent variables boost the performance of function approximation. Secondly, the modulated MLP layer for implicit function parameterization can be further equipped in HNP, referred to as HNP-Mod. Compared with the concatenation of latent variable, using low-dimensional latent variables to modulate the network parameters enables more flexible function approximating. Based on HNP-Mod,

| | number of Modulated MLP blocks | | | | | |
|---|---|---|---|---|---|---|
| | 0 | 2 | 4 | 6 | 8 | 6 / single $\mathbf{z}$ |
| context | 1.375±0.001 | 1.376±0.001 | 1.376±0.001 | 1.376±0.001 | 1.376±0.001 | 1.376±0.001 |
| target | 0.076±0.001 | 0.336±0.028 | 0.371±0.001 | **0.439±0.007** | 0.435±0.028 | 0.245±0.003 |

Table 2: Ablation study about the detailed hierarchical structure on Matern kernel. Here we also provide the results of test log-likelihood (larger is better).

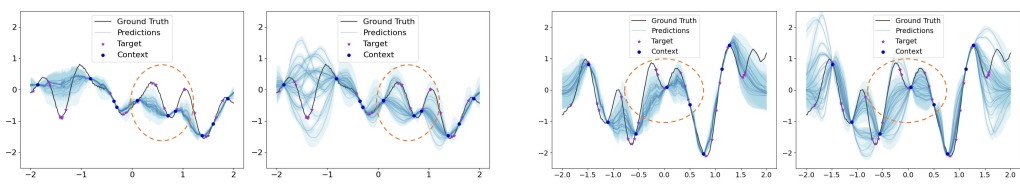

(a) Matern kernel. ANP vs. our VNP.        (b) RBF kernel. ANP vs. our VNP.

Figure 3: Visualizations of the 1D regression results. VNP delivers diverse function prediction results, while ANP (Kim et al., 2019) tends to underestimate the function variances at some target locations.

we finally add the bottleneck encoder to learn conditional network inputs that are adaptive to target locations, building our final model VNP. In Table 1, we can find this powerful pre-processing encoder can further improve the performance, because the set tokenizer (set convolution here) can preserve locality to learn more appropriate features.

In Table 2, we further ablate on the detailed hierarchical structures in our model. As we can observe, when we increase the number of Modulated MLP blocks, the prediction performance (at target points) is improved until saturated with 6 blocks (*i.e.*, $L_K = 6$). In addition, the column with "6 / single $\mathbf{z}$" means we use exactly the same network structures as $L_K = 6$, but instead only impose the KL constraint to the final latent variable $\mathbf{z}_6$. It is found that the performance will also drop dramatically, which verifies that the gains come from the hierarchical design instead of the increased network capacity. This ablation study guides us to set the number of Modulated MLP blocks as 6 to compare with other methods, which can keep a balance between the performance and the complexity.

**Visualizations.** We visualize the obtained function distributions from stacked ANP (Kim et al., 2019) and our VNP, by sampling the latent variables 20 times. Given partial observations of a signal, there exists uncertainty on the continuous function since there are many possible ways to interpret these observations (i.e., context set). An excellent Neural Process model should be able to model such uncertainty through the fitted functions. In other words, the generated functions conditioned on the context set should approximate the data distribution and cover the target set points. As shown in Figure 3, we can see that the generated functions from the stacked ANP cannot predict the target points accurately, *e.g.*, those regions marked by orange. In contrast, our method provide good approximations for the continuous functions, where the distribution covers the groundtruth function.

## 5.2   2D IMAGES COMPLETION AND SUPERRESOLUTION

A 2D image can be modeled by a continuous function that maps the 2D pixel coordinates to the color values. We implement our VNP framework for learning the continuous function of 2D images, which supports tasks such as image completion and super-resolution to arbitrary size.

**Settings.** We conduct experiments on CelebA dataset (Liu et al., 2015), mainly with the resized resolution of $64 \times 64$. Due to the limited representation capability or the high requirement on computation resources, most previous NP methods (Kim et al., 2019; Lee et al., 2020) are in general trained and evaluated on relatively low resolution such as $32 \times 32$. Our framework enables the processing of complex signal with higher resolution. We train a single model that supports image completion and super-resolution tasks. During training, we control the context ratio as 0.03, which means 3% pixels are taken as context set. The target ratio is set as 0.15 and the target set has partial intersection with the context set. More detailed experimental settings can be found in Appendix B.

**Visualizations**. We compare our VNP with stacked ANP (Kim et al., 2019) tested with three different context ratios respectively as shown in Figure 4. Our VNP generates diverse and realistic image completion results with fine details. In comparison, the results of Stacked ANP are blurred.

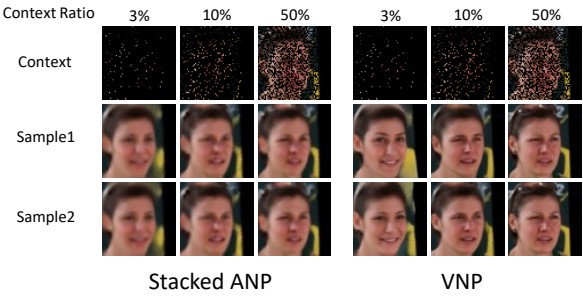

Figure 4: Qualitative comparisons with Stacked ANP (Kim et al., 2019). Our VNP presents diverse and realistic image completion results.

| CelebA64 | NLL (lower is better) context ratio = 0.03 |
|---|---|
| Stacked ANP | 2.988 |
| VNP, $ks = 1$, $L_K = 1$ | 2.994 |
| VNP, $ks = 1$, $L_K = 6$ | 2.953 |
| VNP, $ks = 2$, $L_K = 6$ | 2.952 |
| VNP, $ks = 4$, $L_K = 6$ | 2.964 |

Table 3: Quantitative results measured by Eq.8 on the test set. Lower is better. $ks$ is the kernel size in set tokenizer. $L_K$ is the number of Modulated MLP blocks.

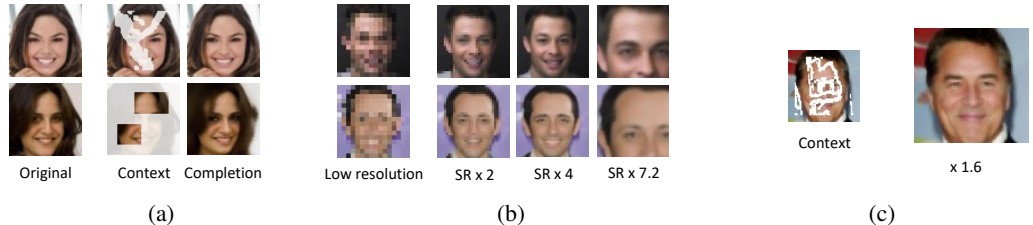

Figure 5: Visualization of VNP results on CelebA64 (Liu et al., 2015). (a) Image Completion. (b) Superresolution to arbitrary size. (c) Handling image completion and superresolution simultaneously.

| GFLOPs | CelebA64 | | | CelebA178 | | |
|---|---|---|---|---|---|---|
| | Context Ratio | | | Context Ratio | | |
| | 0.05 | 0.25 | 1.0 | 0.05 | 0.25 | 1.0 |
| Stacked ANP | 17.1 | 39.8 | 198.5 | 190.0 | 867.2 | 12150 |
| Our VNP | 27.6 | 27.6 | 27.6 | 204.5 | 204.5 | 204.5 |

Table 4: Comparing our VNP with stacked ANP (Kim et al., 2019) about the complexity, measure by FLOPs. The bottleneck encoder plays an important role in reducing the computational cost. The statistic in gray is estimated since we have an upper bound limitation of GPU memory.

In Figure 5, we show that the proposed VNP achieves satisfactory results for image completion and superresolution on CelebA64 dataset. As a unified framework, with a single model, it can support image completion, superresolution, and the concurrent of completion and superresolution.

**Influence of Patch/Kernel Size**. We conduct an ablation study to investigate the influence of the patch/kernel size in our set tokenizer. We set the stride the same as the kernel size. The results are shown in Table 3. It is observed that using larger kernel size does not degrade the performance obviously. In addition, using hierarchical structure brings gains.

**Complexity Comparisons.** Thanks to our bottleneck encoder design, the proposed VNP is a practical framework for complex signal modeling, where the number of context points as condition is usually large. For comparison, we calculate the GFLOPs of stacked ANP and our VNP when target ratio is 0.25 on CelebA64 and CelebA178 respectively. Here, the kernel size of set tokenizer is 4 on CelebA64 and 10 on CelebA178. The computational complexity comparisons in terms of GFLOPs are shown in Table 4. Since we use set tokenizer to reduce the number of tokens and process the context set in image grids, the GFLOPs of VNP would be smaller than that of Stacked ANP, especially when the context ratio is high.

### 5.3 VIEW SYNTHESIS ON 3D SCENE

Implicit neural representations excel at representing 3D signals. However, since 3D signals are usually much more complex, most previous works (Mildenhall et al., 2020; Martin-Brualla et al.,

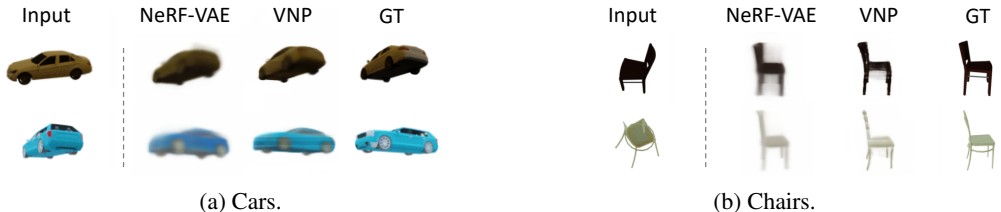

(a) Cars.                                    (b) Chairs.

Figure 6: Novel view synthesis on ShapeNet (Chang et al., 2015) Objects. Our VNP presents more realistic prediction results than the blurry results of NeRF-VAE (Kosiorek et al., 2021).

|  |  | PSNR (dB) | | |
|---|---|---|---|---|
|  |  | Cars | Lamps | Chairs |
| Deterministic | Learned Init (Chen et al., 2021b) | 22.80 | 22.35 | 18.85 |
|  | Trans-INR (Chen & Wang, 2022) | 23.78 | 22.76 | **19.66** |
| Probabilistic | NeRF-VAE (Kosiorek et al., 2021) | 21.79 | 21.58 | 17.15 |
|  | Our VNP | **24.21** | **24.10** | 19.54 |

Table 5: The quantitative results of one-shot novel view synthesis. We compare the proposed VNP with both deterministic and probabilistic methods. All of these methods are able to produce INRs representing the 3D scenes within few steps.

2021) require expensive optimization to fit the neural networks to represent the scenes. We focus on the task of view synthesis in this section to evaluate our proposed VNP.

**Settings.** We follow the spirits of NeRF (Mildenhall et al., 2020) that fits a network to map the world coordinate into the corresponding RGB values and volume density. We use bottleneck encoder (tokenization from the image patches) to calculate the adaptive input of the MLPs, queried by target world coordinates. Then the hierarchically learned global latent variable will modulate the parameters of MLPs to predict the RGB values and volume density. Note that there is an extra volume rendering process inside each decoding block before the pooling module, because it requires transferring the world coordinate to the image coordinate to compute the latent distribution of $\mathbf{z}_k$. We conduct experiments on ShapeNet (Chang et al., 2015) objects, including three sub-datasets: cars, lamps, and chairs. More details on the network structures and hyper parameters can be found in Appendix B.

**Comparisons.** Our VNP model can generate the implicit neural representations *w.r.t.* the previously unseen scene in a single forward pass. In this 3D task, one prior work NeRF-VAE (Kosiorek et al., 2021) can also *generate* the scene from the randomly sampled latent variable. We make quantitative and qualitative comparisons with NeRF-VAE. In addition, we compare with (Chen et al., 2021b) and (Chen & Wang, 2022) which can also learn implicit neural representations of the previously unseen scene, although they are not in the family of probabilistic models and thus cannot model function distributions. As shown in Table 5, our method achieves the best performance in terms of Peak Signal-to-Noise Ratio (PSNR) on Cars and Lamps, and is comparable with Trans-INR (Chen & Wang, 2022) on Chairs. The visualizations in Figure 6a show that VNP produces much better novel-view prediction results than NeRF-VAE, which is also a probabilistic generative model.

## 6    CONCLUSION AND DISCUSSION

The Neural Processes family provides an efficient way for learning implicit neural representations by approximating the function distribution when only partial observation of a signal is given. We propose an efficient NP framework, Versatile Neural Processes (VNP), that largely increases the capability of approximating functions. Our bottleneck encoder and hierarchical latent modulated decoder enables strong modeling capability to the complex signals. Through comprehensive experiments, we show the effectiveness of the proposed VNP on 1D, 2D and 3D signals. Our work demonstrates the potential of neural process as a promising solution for efficient learning of INRs in complex 3D scenes.

ACKNOWLEDGMENTS

This work was supported in part by NSFC under Grant U1908209, 62021001.

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
