# OpenReview forum: "Versatile Neural Processes for Learning Implicit Neural Representations"
_ICLR.cc/2023/Conference — ICLR 2023 poster_

### Official Review · Reviewer_vtKS · 2022-10-20

**Confidence:** 3
**Correctness:** 3
**Technical Novelty And Significance:** 3
**Empirical Novelty And Significance:** 3
**Recommendation:** 8

**Clarity, Quality, Novelty And Reproducibility:**

The paper is clear and well written. It introduces a novel NP architecture by intelligently combining previous work from the literature (ANP, set tokenizer and modulated layers).

The paper provides details about the model structure and experimental setup which seem sufficient to reproduce similar experiments, but the exact structure of the models (number and size of layers, etc..) are not given, even in appendix.

**Strength And Weaknesses:**

**Strengths:**
- The proposed architecture and its training procedure is described with extensive details.
- The proposed VNP is empirically compared to many other models from the literature on various tasks.
  - The improvement on 1D tasks are impressive, showing how VNP is able to capture the variability of the function distribution much more effectively than previous models.
  - The comparison of computational cost on the image reconstruction task clearly shows the gain from the added encoder, making the prediction cost almost independent on the input size.

**Weaknesses:**
- The tasks used to asses the model quality are artificial problems. This is an issue shared by the other NP models from the literature, but as the model family is maturing it would be interesting to asses its applicability to real world problems. After reading the paper I don't have a clear view of when a practitioner would want to use a VNP.
  - In particular, in the context of the super-resolution task, VNP is compared to other NP models, but not to the general state of the art of super-resolution models, which also contains models able to do arbitrary-scale super-resolution (such as *Learning Continuous Image Representation with Local Implicit Image Function*, Chen, Liu & Wang, 2021, cited in the paper).

**Summary Of The Paper:**

This paper proposes a new architecture for Neural Processes named Versatile Neural Process (VNP). This new architecture is build as an improvement over previous Attentive Neural Process (ANP) with two main modifications:
1. The context points are pre-processed by a encoder module built using set convolutions and self-attention layers. This module summarizes the (potentially large) set of context points into a much smaller set of context tokens. This allows the model to scale to much larger sets of context points than ANP.
2. The decoder part of the NP is augmented with a hierarchy of random variables (as shown by (Child 2021), this has a large impact on expressiveness to VAE-like models). These variables are used to module the prediction in an iterative way.

These two modifications together are empirically shown to significantly improve the capacity of VNP to model complex and varied functional distributions, as well as computationally scale to large inputs such as super-resolution tasks.

**Summary Of The Review:**

This paper introduces a new iteration on the design of NPs which significantly improves the capacity of the model to capture variability in the function distribution as well as its scalability with regard to the number of context points.

While the application cases for NPs remains rather nebulous to me, I believe this is a good paper.

---

> ### Author Response · Authors · 2022-11-17
> **Response to Reviewer vtKS**
>
> We thank the reviewer for the thoughtful review.
>
> Q1: I don't have a clear view of when a practitioner would want to use a VNP/NP.
> A1: Thank you for raising this thoughtful comment. As the Neural Process family is maturing, the research community could explore its applicability to real world problems. The Neural Process facilitates fast inference of function approximation rather than leveraging iterative optimization to fit a signal in inference. Moreover, the uncertainty awareness makes it more naturally model the physical world. It has great potential to be applied in many real world problems. Actually, there is a track of research that leverages the existing Neural Process models to deal with domain-specific problems. Neural Processes have been successfully employed to solve problems from multiple perspectives: recommender systems, hyperparameter optimization, neuroscience, space science, and physics-informed modeling of dynamic systems (Jha et al., 2022). These works shed some lights on the applicability of NP/VNP on real world problems.
> Even though there is great progress on Neural Processes and our VNP outperforms other Neural Process methods, when compared with the task-specific designed models, such as the state-of-the-art super-resolution model (Liu et al., 2021), our VNP is still inferior to them. More efforts need to be make to remedy such gap in the future.
>
> Q2: More details on the exact structure of the models.
> A2: We have added more details in our Appendix A and B. We will release our source code after cleaning.

---

### Official Review · Reviewer_HSYx · 2022-10-21

**Confidence:** 3
**Correctness:** 3
**Technical Novelty And Significance:** 2
**Empirical Novelty And Significance:** 2
**Recommendation:** 6

**Clarity, Quality, Novelty And Reproducibility:**

Generally, this paper is easy to follow with clear descriptions. However, as for me, it is limited in technical novelty and somewhat looks like an incremental revision over existing approaches. Please see comments above.

**Strength And Weaknesses:**

Strengths

(1) This paper is generally well-written and easy to follow, except some points. I can understand most statements easily. For some improvement suggestions, please see the weakness part below.

(2) The authors present consistly improved results over chosen baselines under three different tasks.

(3) Most of the technical elements are built on existing literatures, providing solid effectiveness foundations.

Weakness

(1) The novelty of this submission seems to be limited. For example, using self-attention layers && set tokenizer (in Sec. 4.1) and hierarchical latent codes (in Sec. 4.2) to improve results is not new. Although the combinations of these strategies produce appealing results, the authors do not show specific design insights, making the proposed method look a bit ad-doc.

(2) There are no discussions on the limitation and failure cases.

(3) The current developed framework is more complex than previous methods. Given some practical improvements, I am curious about how much time it needs for the training and testing? Please report the training and inference speed as well as corresponding experimental environment for more comprehensive evaluations.

(4) For the baseline comparisons, why do not the authors make evaluations over optimization-based implicit representation modeling methods (e.g. SIREN, Fourier Feature Network)? I look forward to seeing these empirical comparision results.

(5) In terms of experimental results, the used datasets are too simple. How about the results produced by NeRF's synthetic datasets, the complex shape sets used in SIREN and NGLoD, et al.? Please use more complex datasets to prove the technical advantages.

(6) Some minor presentation issues:

A. In Sec. 4.2, the functionality of cross-attention is unclear. Why should we use the cross-attention here?

B. In Sec. 4.2, why multiplying the KL term with a small weight can better capture the uncertainty of function distributions?

**Summary Of The Paper:**

This paper presents an auto-encoder based pipeline to achieve versatile neural process, which consists of a bottleneck encoder and modulated MLP based decoder. The bottleneck encoder aims to produce fewer and informative context tokens while the decoder hierarchically learns multiple global latent variables and the uncertainty of a function to improve the capability in modeling complex signals. While the developed method follows the existing paradigms quite a bit, it shows practical improvements compared to chosen baselines under three different tasks.

**Summary Of The Review:**

Although this paper yields some interesting empirical results compared to some baselines, the proposed method builds mainly on top of previous techniques without enough technical contributions. Additionally, it is also weak in the experimental part, including lack of evaluations over challenging datasets and optimization-based methods. However, I am not an expert in this area, so I will not object if the other reviewers tend to accept and think it has already met the bar of ICLR.

---

> ### Author Response · Authors · 2022-11-17
> **Response to Reviewer HSYx (Part One)**
>
> We thank the reviewer for the detailed review and constructive feedback towards improving our manuscript.
>
> Q1: Clarification of the novelty.
> A1: The core contribution of our paper is that we propose a new Neural Process method, VNP, which significantly advances the development of Neural Processes. Previous NPs either have inferior modeling capability (e.g., NP, ConvNP) or suffer from high computation costs (e.g., ANP). We address these by (a) designing a hierarchical structure with multiple latent variables that empowers NP to have high modeling capability, and (b) introducing set tokenizer that significantly alleviates computational complexity. With the merits of NPs (i.e., fast inference of implicit neural representation for unseen signal, probabilistic modeling), our VNP has advanced a significant step forwards the applicability of NPs to practical problems.
>
>
> Q2: There are no discussions on the limitation and failure cases.
> A2: Thanks for the suggestion. We have added such discussion in Appendix D.
> The proposed VNP inherits the limitation of NP family. As referred by Reviewer L3M4, NPs are interesting techniques for meta-learning implicit neural representations due to their reduction of the high cost of training. NPs learns the common knowledge shared by the dataset, enabling fast inference of an unseen signal without the need of finetuning. NPs still cannot work well for dataset including diverse objects (with less shared knowledge), e.g., ImageNet.
>
> Q3: Please report the training and inference speed and the corresponding experimental environment.
> A3: We have added the speed in Appendix in the revision. All the experiments are performed on V100 (1 V100 for 1D, 4 V100 for 2D and 3D experiments).
> Specifically, training a VNP model for the 1D regression task requires about 5 hours. Training a 2D VNP model on CelebA64 requires about 24 hours. Training a 3D VNP model for novel view synthesis requires about 40 hours.
> For the inference speed, VNP requires very short time for 1D and 2D tasks. On 1D regression task, VNP takes 0.285 second for testing a batch with batch size of 2000. On the CelebA64 dataset (2D task), our VNP takes around 0.112 second to infer a single batch of size 8. On the 3D Cars dataset, our VNP takes 5.28 second to render an image with a novel view (reasons explained in Appendix C). All these results are tested with a single V100 GPU.
>
> Q4: For the baseline comparisons, why do not the authors make evaluations over optimization-based implicit representation modeling methods (e.g. SIREN, Fourier Feature Network)? I look forward to seeing these empirical comparison results.
> A4: Many thanks for your suggestion. We have updated our paper to include this group of experiments in Appendix C. In general, optimization-based implicit representation modeling methods require many iterations of optimization to fit a signal in testing, which is computationally costly. In contrast, NPs facilitates fast inference of a signal without requiring further optimization in testing.
> We implement SIREN for the task of novel view synthesis from a single view image, with the same experimental setting of our method. On the 3D ShapeNet dataset (the category of Cars), we obtain the statistics of prediction performance (measured by PSNR) vs. the iteration steps. The results are shown below.
>
> |                   |  SIREN            |  SIREN            |  SIREN            |  SIREN            |  SIREN            |  VNP              |
> |-------------------|-------------------|-------------------|-------------------|-------------------|-------------------|-------------------|
> | Iteration Number  |    1              |     3             |    30             |    100            |    300            | no iteration during testing      |
> | Times (s)         |    0.21           |     1.35          |    3.68           |    11.59          |    33.68          | 5.28              |
> | PSNR (dB)         |    11.92          |     12.27         |    12.72          |    12.73          |    12.00          | 24.52             |
>
>
>
> We can see that given only a single view image as context, our method delivers much better prediction performance than SIREN, even if SIREN is optimized to fit for the signal with many iterations. This attributes to that our VNP can learn prior of the data distribution to complete the representation of a signal. On the other hand, our method does not need finetuning in inference.

---

> > ### Comment · Reviewer_HSYx · 2022-11-22
> > **Response to authors**
> >
> > Hi authors,
> >
> > Thank you for your detailed response to my questions. I think most of my concerns have been addressed so I raised my score to 6.

---

> ### Author Response · Authors · 2022-11-17
> **Response to Reviewer HSYx (Part Two)**
>
> Q5: Experimental results on more complex dataset.
> A5: Thanks for the suggestion. For the 3D experiment, following Trans-INR (Chen & Wang, 2022), we conduct experiments on the ShapeNet dataset. It contains the simple categories of object, such as Chairs and Cars. Our VNP can outperform the previous work including Trans-INR and NeRF-VAE (Kosiorek et al., 2021) with the same experimental settings. However, the generated results are still somewhat blurry. Even though we have made significant progress for NPs, there is still large space to advance the techniques to make them work excellently on complex dataset.
>
>
> Q6: Why should we use the cross-attention here?
> A6: The cross-attention is used for generating the coarse prediction at each target location, by leveraging the all known context tokens (as key and value). This is a operator originally proposed by Attentive Neural Processes.
>
> Q7: Why multiplying the KL term with a small weight can better capture the uncertainty of function?
> A7: Our loss objective contains two terms, the reconstruction term and the KL divergence term, with the weight to balance the importance of the two terms. If we multiply a small weight $\beta$ upon the KL term, there is less constraint on the distribution alignment between the prediction and ground truth, which will thus make our prediction results more precise. Note that given partial observations of a signal, there is uncertainty of the functions. This strategy is thus used to model the uncertainty while ensuring the reconstruction quality of the context set.

---

### Official Review · Reviewer_L3M4 · 2022-10-22

**Confidence:** 3
**Correctness:** 3
**Technical Novelty And Significance:** 3
**Empirical Novelty And Significance:** 3
**Recommendation:** 6

**Clarity, Quality, Novelty And Reproducibility:**

`Clarity & Quality:`
The clarity of the submitted manuscript is lacking, requiring the reader to consult several cited works to understand key contributions. Technical language is sometimes imprecise, using existing terms in unusual fashion or referring to certain related works by alternative names. Grammatical errors exist. Mathematical terms or unclear, notation is inconsistent. Examples:

- Section 4.2: "we use the modulated fully-connected (ModFC) layer (Karras et al., 2020; 2021) to adjust the inherent MLPs parameters". The authors should state the equations for those layers, saving the reader the effort to look this up. What are "inherent" MLP parameters?

- (Garnelo et al., 2018b) should be referred simply as "Neural Processes" instead of "Latent Neural Processes", likewise (Garnelo et al., 2018a) is a "Conditional Neural Process".

- "ANP has troubles in processing complex signals that requires abundant context points as condition (e.g., image with high resolution), where the computational cost is very expensive." - Statements like this should be made more precise my stating the computational complexity of the ANP attention mechanisms (provided in those works). The proposed method should be compared in terms of computational complexity.

- Figure 1 is difficult to parse when encountered for the first time. For a first schematic overview of the model there is too much detail, which can be omitted given that Figure 2 shows the architecture in more detail anyway.

- Section 4.1: "can reduce the number of tokens by 100 times". What is the image size in question?

- Equation (3): $q_{\phi}(\mathbf{z}|D_T)$ is not defined (merely called "conditional posterior encoder") - needs more explanation, $z$ is not bold in the sentence before.

- Equation (6): The sum is over $i$, while $i$ does not appear inside the summation.

- Equation (7): $z$ is again not bold, causing confusion (especially since $\mathbf{z}$ is redefined just before Equation (6)). Also, shouldn't the likelihood term also includes sampling and conditioning from all latent variable $\mathbf{z}_{1:K}$ used in the decoder?

- Introduction: "Moreover, for complex signal, [...]" -> "complex signals", or "a complex signal"

- Section 4.1: "the computational burden is heavy and such framework is impractical" -> "such a framework".

- Section 4.2: "After achieving the low-dimensional latent variable" -> What does it mean to "achieve" a latent variable? Do the authors mean "sample"?

- Section 4.2: "they learn a single global" -> omit using "they"

- Figure 3 caption: "tends to underfit the functions at some target locations" -> "underfitting" is usually a term reserved for context points in the NP literature. The graphic shows clearly this does not in fact happen, indeed the ANP shows better context set reconstruction (Table 1). The authors most likely mean "underestimates the variance".

`Reproducibility:`
    - Difficult to make a precise statement here without a concrete attempt. I'd urge the authors to provide an implementation in the supplementary mterial

**Strength And Weaknesses:**

`Strengths:`
- Neural Processes are interesting techniques for meta-learning INRs due to their reduction of the high memory cost of training, making this an interesting topic to investigate. Reducing the computational complexity of the ANP's self-attention is a valuable contribution.
- Results seem overall strong (Table 1, Figure 3), fewer FLOPs are required as stated (Table 3).

`Weaknesses:`
- While the Set Tokenizer is shown as a key innovation in the presented work, reducing the complexity of the attention mechanism, little insight is provided in terms of the cost that is paid by the summarisation of multiple context points. Table 1 shows that as expected, context reconstruction quality does decrease, but it is unclear how that decrease depends on the kernel size and stride (Section 4.1). No complexity analysis (in terms of $\mathcal{O}$ notation) is provided.
- Due to a large number of architectural and algorithmic changes, the authors introduce a large number of additional hyperparameters to be tuned by a partitioner (number of hierarchical blocks/number of latent variables, $\beta$ (KL objective), kernel width/stride of set transformer). Reading between the lines, the method appears to suffer from potential instability "We only modulate some of the MLPs to preserve the training stability."
- The manuscript in its current form has several clarity and quality issues making it currently unsuitable for publication. The submission appears rushed and with insufficient attention to writing.

`Questions:`
- To which extent are the improved results in Table 1 explained by the significantly larger models? What happens when baseline architectures are scaled to provide an equal comparison in terms of #parameters?

**Summary Of The Paper:**

The authors present new architectural improvements to the Neural Process family, arguing that the costly self-attention mechanism designed to overcome context underfitting introduced in the work on Attentive Neural Processes (Kim et al., 2019) makes the method family unsuitable for large signals. Instead, the authors propose the use of a set tokenizer (Zaheer et al., 2017) which summaries several neighbouring context points into individual tokens, does reducing the complexity of the ANP attention mechanism. A second innovation are significant architectural changes to the decoder, introducing hierarchical latent variables, modulated MLP blocks as well as other changes. The authors evaluate their method on NP tasks and compare to a wide variety of other NP architectures.

**Summary Of The Review:**

Overall an interesting idea with the potential for publication, but currently unsuitable due to serious Clarity & Quality issues. Will consider raising my score if the authors significantly improve those aspects during a rebuttal.

---

> ### Author Response · Authors · 2022-11-17
> **Response to Reviewer L3M4 (Part One)**
>
> We thank the reviewer for the careful reading and constructive feedback towards improving our manuscript. We address the reviewer's points below and have updated our manuscript accordingly.
>
> Q1: Provide complexity analysis (in terms of $\mathcal{O}$ notation).
> A1: Thank you for the nice suggestion. Adding this can help the readers to have a better understanding of the effects of the set tokenizer design. We have added the analysis in Section 4.1 in our revision.
> Taking a 2D image signal as an example, with the kernel size of $k \times k$ and stride of $k$, the set tokenizer reduces the number of tokens by $k^2$ times (assume the image resolution is an integer multiple of $k$). By using the set tokenizer, the complexity of attention layers is reduced from $\mathcal{O}(L_sN_C^2 + L_cN_CN_T)$ to $\mathcal{O}(L_s\frac{N_C^2}{k^4} + L_c\frac{N_C}{k^2}N_T)$, where $L_s$ and $L_c$ are the number of the self-attention layers and the cross-attention layers, $N_C$ and $N_T$ are the number of context points and target points, respectively.
>
> Q2: The influence of the kernel size.
> A2: Thanks for your suggestion. In the revision, we have added the analysis about the the influence of kernel size/stride on the celeba64 dataset in Table 3. For simplicity, we set the stride the same as the kernel size (i.e., no overlap) in the set tokenizer. The experimental results are shown below:
>
> |  CelebA64                 | Loss (lower is better) |
> |---------------------------|------------------------|
> | Stacked ANP (pixel-wise)  |    2.988               |
> | VNP, ks = 1, L_K = 1      |    2.994               |
> | VNP, ks = 1, L_K = 6      |    2.953               |
> | VNP, ks = 2, L_K = 6      |    2.952               |
> | VNP, ks = 4, L_K = 6      |    2.964               |
>
> Here, ks denotes the kernel size in set tokenizer, and $L_K$ denotes the number of modulated MLP blocks. We can see that using larger kernel size does not degrade the performance obviously. In addition, using  hierarchical structure brings gains.
>
> Q3: To which extent are the improved results in Table 1 explained by the significantly larger models? What happens when baseline architectures are scaled to provide an equal comparison in terms of #parameters?
> A3: Thanks for the suggestion. We have conducted an additional group of experiments to make a fair comparison, where we enhance baseline model Stacked ANP to make it have similar model size with our VNP by increasing the number of channels. The results shown below demonstrate that the performance of stacked ANP with a larger model size is only slightly improved and is still inferior to our VNP. We have added the results in Table 1 in our revised manuscript.
>
> |  Matern Kernel       | Target Log Likelihood | Model Size |
> |----------------------|-----------------------|------------|
> | Stacked ANP          |    0.183 $\pm$ 0.017  | 1.52M      |
> | Enhanced Stacked ANP |    0.188 $\pm$ 0.016  | 2.31M      |
> | Our ANP              |    0.439 $\pm$ 0.007  | 2.29M      |

---

> > ### Comment · Reviewer_L3M4 · 2022-11-18
> > **Response to authors**
> >
> > Dear authors,
> >
> > Thank you for the detailed response and for improving the typos and clarity of the manuscript. As suggested, I'm happy to raise my score to 6.

---

> ### Author Response · Authors · 2022-11-17
> **Response to Reviewer L3M4 (Part Two)**
>
> Q4: Improve the clarity, preciseness of writing.
> A4: We appreciate the thoughtful review and nice suggestions for improving the paper. We have carefully revised our paper accordingly and response each comment/question below.
>
> (1) "Section 4.2: "we use the modulated". The authors should state the equations for those layers,... What are "inherent" MLP parameters?
> We have added details in Appendix A to introduce the mechanism of modulated MLP. We removed the word "inherent" to be precise.
> (2) "（Garnelo et al. 2018b）should be referred simply as "Neural Processes" instead of "Latent Neural Processes"".
> Yes, we have modified them correspondingly.
> (3) ""ANP has troubles in ..." ... The proposed method should be compared in terms of computational complexity."
> We have added the computational complexity analysis in terms of  $\mathcal{O}$ notation around Eq. (4) in our revision (see the answer to Q1). In Table 3, the comparison in terms of FLOPs also shows the advantage of our VNP over Stacked ANP.
> (4) "Figure 1 is difficult to parse when encountered for the first time..."
> We follow your good suggestion and have modified Figure 1 to a simplified version.
> (5) "Section 4.1: 'can reduce the number of tokens by 100 times'. What is the image size in question?"
> We have revised that to be more precise as: "Taking a 2D image as an example, with the kernel size of $k \times k$ and stride of $k$, the set tokenizer can reduce the number of sample points by $k^2$ times (assume the image resolution is an integer multiple of $k$)." For example, for an image of 178 $ \times $ 178, after padding, we have an image of 180 $ \times $ 180. When k=10 (where a token corresponds to a patch of size 10 $ \times $ 10=100), the number of tokens after the tokenization is thus 18 $ \times $ 18.
> (6) "Equation (3): $q_{\phi}(\mathbf{z}|D_T)$ is not defined (merely called "conditional posterior encoder") - needs more explanation, z is not bold in the sentence before."
> We have added explanation: $q_{\phi}(\mathbf{z}|D_T)$ denotes the posterior distribution of the latent variable given the ground truth target points, which is only used in training and not accessed in inference. We have corrected the errors.
> (7) "Equation (6): The sum is over i, while i does not appear inside the summation."
> We have corrected  Eq. (6) (now is Eq. (7)) by modifying "i=k" as "k=1".
> (8) "Equation (7): z is again not bold, ... Also, shouldn't the likelihood term also includes sampling and conditioning from all latent variable z1:K used in the decoder?
> We have corrected Eq. (7) to make likelihood term include all latent variable z1:K.
> (9-13) We have fixed all the typos and writing issues you mentioned.
>
> For reproducibility, we have submitted the code for the 1D experiments in the updated supplementary material. We will release all the other code after necessary cleaning.

---

### Official Review · Reviewer_vL4F · 2022-11-04

**Confidence:** 3
**Correctness:** 4
**Technical Novelty And Significance:** 3
**Empirical Novelty And Significance:** 3
**Recommendation:** 8

**Clarity, Quality, Novelty And Reproducibility:**

The paper is technically sound and novel. It is generally well written(except for some minor issues mentioned above) and easy to follow. It will be more reproducible if the authors can release the code and data.

**Strength And Weaknesses:**

**Strength**
1. The proposed encoder structure is efficient in modeling complex signals especially for 2D and 3D data, compared with previous works (Attentive NP, Transformer NP, ConvNP, etc.)
2. The authors conduct extensive experiments on 1D, 2D and 3D data separately to demonstrate the effectiveness of the proposed framework on modeling different signals, compared with previous SOTA methods.
3. The authors also perform sufficient ablation studies to validate the design choices.

**Questions**
1. In fig.3 the results of VNP are pretty diverse, does this mean that the prediction is unstable/noisy?
2. For image completion comparison, are you using the same resolution(64 or 32?) for a fair comparison? Does the blurred results come from low-resolutions instead of a inferior method?

**Writing**
1. Typo: fig.4 "Our HINP"
2. I would suggest to ensure the tab/fig is on the same page as the text that mentions them. This will make reading easier. For example, fig.2, tab.2, fig.4.

**Summary Of The Paper:**

This paper proposes an encoder-decoder architecture with strong modeling capability on complex signals, which is capable of capturing the distribution of 1D, 2D and 3D signals efficiently. The main contributions are:
1. a framework that increases the capability of approximating complex signals (from 1D signal to 2D images and 3D shapes).
2. an efficient encoder that can significantly reduce computational cost.
3. a hierarchical decoder that learns multiple global latent variables for better approximation of global structure of a continuous function.

**Summary Of The Review:**

Given the above-mentioned strength (effectiveness over previous methods, extensive analysis, etc), my suggestion is that the paper is good for a publication.

---

> ### Author Response · Authors · 2022-11-17
> **Response to Reviewer vL4F**
>
> We thank the reviewer for the valuable feedback and constructive suggestions! We address the reviewer's points below.
>
> Q1: In fig.3 the results of VNP are pretty diverse, does this mean that the prediction is unstable/noisy?
> A1: Actually, the diverse results do not mean our predictions are unstable/noisy. Instead, they demonstrate the superior modeling capability of our method. We have added more explanations in our revision to avoid confusion. Given partial observations of a signal, there exists uncertainty on the continuous function (which represents this signal) since there are many possible ways to interpret these observations (i.e., the context set). An excellent Neural Process model should be able to model such uncertainty of the functions. In other words, the generated functions conditioned on the context set should be able to approximate the data distribution that covers many possibilities of the target set points (where the target set can be seen as samples located on one instantiation of the function). In fig.3, we can see that some sampled functions from our model are able to cover the target points, while the previous method ANP usually misses some target points. This demonstrates the superior uncertainty modeling capability of our method. In addition, the quantitative comparisons in Table 1 also show the superiority of our models.
>
>
> Q2: For image completion comparison, are you using the same resolution(64 or 32?) for a fair comparison? Does the blurred results come from low-resolutions instead of a inferior method?
> A2: We use the same resolution (64 $\times$ 64) for both Stacked ANP and our VNP. The results of Stacked ANP are more blurry than that of our VNP. The superiority of our model attributes to the using of multiple latent variables to jointly model the function distributions, enabling stronger modeling capability.
>
> Q3: Typo and paper layout.
> A3: Thank you very much for pointing out these and providing nice suggestions. We have adjusted the positions of tabs/figs to make them close to the related text.
>
>
> We have submitted the code for the 1D experiments in the updated supplementary material. We will release all the code after necessary cleaning.

---

> > ### Comment · Reviewer_vL4F · 2022-11-30
> > **Response to authors**
> >
> > Thank you for the detailed explanation and clarification. I will keep my score as 8.

---

### Public Comment · ~Richard_S_Liu1 · 2022-11-15
**Equations of generative processes and the Correctness of derived ELBO?**

Hi Authors,

Thanks for the work. I take great interest in the hierarchical modeling of neural processes. So I have a couple of questions about this work.

1. In the vanilla neural process, it gives the generative process as $p(y_{1:n}|x_{1:n})=\int p(z)\prod_{i=1}^{n}p(y_i|x_i,z)dz$. ***Can you specify the generative process of your proposed hierarchical model in equations?*** Meanwhile, ***does your hierarchical neural process meet the Kolmogorov extension theorem to verify the definition of exchangeable stochastic processes?***
2. Since the sequential latent variables $z_{1:L}$ are correlated and factorized in hierarchical variational autoencoders [1], so the derived evidence lower bound has the KL divergence term in the form $\sum_{l=2}^{L} E_{q(z_{l}|z_{<l},x)}[KL(q(z_{l}|x,z_{<l})||p(z_{l}|z_{<l}))]+KL(q(z_{1}|x)|| p(z_1))$.
***I noticed the expectation form and the correlation are removed in your ELBO equation (6). So is this a correct optimization objective, and can you provide detailed math formulations?***
3. Is it possible to attach the link to the anonymous code for implementation details?

***Reference***

[1] Vahdat, Arash, and Jan Kautz. "NVAE: A deep hierarchical variational autoencoder." Advances in Neural Information Processing Systems 33 (2020): 19667-19679.

---

> ### Author Response · Authors · 2022-11-18
> **Response to Richard S Liu**
>
> Thank you for your interest to this work and comments for improving our paper.
>
> Q1: Can you specify the generative process of your proposed hierarchical model in equations? Meanwhile, does your hierarchical neural process meet the Kolmogorov extension theorem to verify the definition of exchangeable stochastic processes?
>
> A1: VNP can still be formulated as
>
> $p(Y_T|X_T, D_C) = \int p(Y_T|X_T, D_C) p(\mathbf{z}|X_T, D_C) d\mathbf{z},$
>
> where $p(\mathbf{z}|X_T, D_C)=p(\mathbf{z}_1|X_T, D_C)p(\mathbf{z}_2|\mathbf{z}_1, X_T, D_C) ... p(\mathbf{z} _{L_K}|\mathbf{z} _{<L_K}, X_T, D_C)$.
>
> This is similar to the generative process of hierarchical VAEs, such as NVAE and very deep VAE (VDVAE).
>
>
> The proposed VNP meets the Kolmogorov extension theorem. VNP is permutation-invariant w.r.t. the order of sample points (exchangeability), where each of our latent variables (in our hierarchical structure) aggregates the sample information by average pooling. This is a special design in the proposed hierarchical neural process, different from the previous hierarchical VAEs.
>
> Q2: Is Eq (6) a correct optimization objective, and can you provide detailed math formulations?
>
> A2: (1) We missed the expectation form in Eq (6) (now Eq. (7)). Many thanks for your question. We have corrected this equation in our revision as
>
> $ D _{KL} = \sum _{k=2}^{L_K} \mathbb{E} _{q _{\phi}(\mathbf{z} _{<k}| D_C, D_T)} [ D _{KL} [q _{\phi}(\mathbf{z} _k|\hat{Y} _{<k}, D_C, D_T) || p _{\psi}(\mathbf{z} _k|\hat{Y} _{<k}, D_C, X_T)]] + D _{KL} [q _{\phi}(\mathbf{z}_1|D_C, D_T) || p _{\psi}(\mathbf{z}_1|D_C, X_T)], $
>
> where $q _{\phi}(\mathbf{z} _{<k}|D_C, D_T) = \prod _{i=1}^{k-1} q _{\phi}(\mathbf{z} _{i}|\hat{Y} _{<i}, D_C, D_T)$ is the approximate posterior of $\mathbf{z} _{<k}$.
>
>
> (2) The correlation of latent variables is preserved in this equation.
>
> Note that $ \hat{Y} _ k $ here is determined by $ \mathbf{z} _ { \leq k } $  (and $X_T, D_C$),
> since the parameters of the modules to infer $Y _ k$ are modulated by $ \mathbf{z}  _ {\leq k} $ (see Figure 2). That is to say,
> $\hat{Y}  _ k \leftarrow (\mathbf{z} _ {\leq k}, D_C, X_T), k>0$.
> Therefore, Eq (6) (now Eq. (7)) in our paper is equivalent to
> $ D _ {KL} = \sum _ {k=2}^{L _ K} \mathbb{E} _ {q _ {\phi}(\mathbf{z} _ {<k}| D_C, D_T)} D _ {KL} [q _ {\phi}(\mathbf{z} _ k|\mathbf{z} _ {<k}, D_C, D_T) || p_{\psi}(\mathbf{z} _ k|\mathbf{z} _ {<k}, D_C, X_T)] + D _ {KL} [q _ {\phi}(\mathbf{z} _ 1|D_C, D_T) || p _ {\psi}(\mathbf{z} _ 1|D_C, X_T)]$,  which preserves the correlation of the latent variables.
>
> Thanks for your pointing out it.
>
> Our code for implementing the 1D regression task is attached in the supplementary material (without in-depth cleaning). Other code will be released after necessary cleaning.

---

### Author Response · Authors · 2022-11-17
**Rebuttal Summary**

We appreciate the positive comments and constructive suggestions from all the reviewers.
We have revised our manuscript following the suggestions, with the major modifications marked by blue. We hope our following responses address all the concerns. Welcome to raise questions if you have.

Correction:
For the evaluation of target log likelihood on 1D regression task, we found there is a bug in our code. In the revision, we have updated the numerical results after fixing this bug. The trends on performance are the same as our previous version. We are sorry for this mistake.  We have submitted the code of 1D regression experiment for reproducibility. The code of 2D and 3D experiments will be released by necessary cleaning.

---

### Public Comment · ~Richard_S_Liu1 · 2022-11-18
**Mathematically incorrect ELBO (Optimization Objective) and Incorrect Evaluation in Experiments (Code Implementations)?**

Hi Author and Other Reviewers,

I double-checked the revised optimization objective and the uploaded codes and found crucial points incorrect in the manuscript.

***1. Incorrect ELBO***

**The evidence lower bounds (ELBOs) as the model optimization objective in the latest three versions of the manuscript are different from each other, and none of them are correct in math**. Note that the KL divergence term inside the Eq. (7) should be $\sum_{l=2}^{L} E_{q_{\phi}(z_{l}|z_{<l},x)}[KL(q_{\phi}(z_{l}|x,z_{<l},D_{C},D_{T})||p_{\psi}(z_{l}|z_{<l},D_{C},X_{T}))]$. This can be verified in step-by-step derivations.
Meanwhile, the used approximate posterior **is not hierarchical at all, since $q_{\phi}(z_{<k}|D_C , D_T ) = \prod_{i=1}^{k-1} q_{\phi}(z_{i}|Y_{<i},D_C,D_T)$ assumes the independence of all $z_i$ (these l.v.s are not correlated at all in the paper)**.


***2. Incorrect Evaluation (Importance-Weighted ELBO? in testing from Uploaded Codes)***

Take a look at the uploaded code in the catalogue **"…/regression/model/vnp.py"**, the Class VNP forward section: when else (not self.training), **it computes the ELBO  and then uses the elbo to obtain the evaluation result from the $ll = logmeanexp(loss)$**. This is different from the evaluation in NP models, which use the importance-weighted likelihoods instead of ELBO.

---

> ### Author Response · Authors · 2022-11-18
> **Response to Richard S Liu**
>
> Hi, Richard, please see our responses below.
> Q1: Incorrect ELBO?
> A1: As shown in our responses to your previous questions, $\hat {Y} _ {\leq k}$ is determined by $z _ {\leq k}$, which can be formulated as $\hat{Y}  _ k \leftarrow (\mathbf{z} _ {\leq k}, D_C, X_T), k>0$. The prediction of the next latent variable $\mathbf{z} _ { k} $ is based on $\hat {Y} _ {\leq k-1}$, thus based on $\mathbf{z} _ {\leq k-1}$. Therefore, the latent variables $\mathbf{z} _ k, k = 1, 2, ..., L_K,$ are not independent. And Eq. (7) in our paper is equivalent to $ D _ {KL} = \sum _ {k=2}^{L _ K} \mathbb{E} _ {q _ {\phi}(\mathbf{z} _ {<k}| D_C, D_T)} D _ {KL} [q _ {\phi}(\mathbf{z} _ k|\mathbf{z} _ {<k}, D_C, D_T) || p_{\psi}(\mathbf{z} _ k|\mathbf{z} _ {<k}, D_C, X_T)] + D _ {KL} [q _ {\phi}(\mathbf{z} _ 1|D_C, D_T) || p _ {\psi}(\mathbf{z} _ 1|D_C, X_T)].$  (by replacing $\hat {Y} _ {\leq k}$ to $\mathbf{z} _ {\leq k}$).
> Therefore, it is a correct hierarchical design.
>
> Q2: Incorrect Evaluation (Importance-Weighted ELBO)?
> A2: The importance weighted ELBO used here is still a valid metric. It has been proved in Importance Weighted VAE [1] that the gap between the ELBO and the data log likelihood will be negligible if there is enough number of samples. During testing, for every group of context set, we will sample the latent variable for many times from the posterior distribution (eval_num_samples=100), and then calculate the lower bound of each sample (note that during testing, kld = (qz.log_prob(z) - pz.log_prob(z)).sum(-1) in our code). We report the final log likelihood after importance reweighting.
>
> Using this metric here is reasonable. For hierarchical latent variable models, it is impossible to sample enough number of latent variable from prior distribution to calculate the importance-weighted likelihoods, because the hierarchical latent variables should be sequentially sampled. For example, if a single latent variable model should sample 100 times, then in a hierarchical latent variable model with 6 blocks, we should sample $100^6$ times from the prior distribution. Therefore, we have to switch to another valid metric to measure the performance of our hierarchical VNP model. In fact, previous work involving hierarchical latent variables such as NVAE also uses this metric to test the log likelihood (https://github.com/NVlabs/NVAE).
>
> In addition, in Table 1, the results of both ANP and our VNP are tested with this metric. Our method achieves obviously better results compared with Stacked ANP, with exactly the same test condition.
>
> [1] Importance Weighted Autoencoders. Burda et al., ICLR2016.

---

### Decision · Program_Chairs · 2023-01-20

**Decision:**

Accept: poster

**Justification For Why Not Higher Score:**

There are typos and the clarify of the paper needs improvement.

**Justification For Why Not Lower Score:**

N/A

**Metareview: Summary, Strengths And Weaknesses:**

This paper introduces a new architecture for versatile neural process. The new approach shows practical improvements over baselines. The reviewers generally agree that the method is interesting and convincing, and the results are strong. The AC agrees with the reviewers and recommends accepting the paper.

**Note From Pc:**

if the above contains the word "oral" or "spotlight" please see: "oral" presentation means -> notable-top-5% and "spotlight" means -> notable-top-25%. As stated in our emails, we are disassociating presentation type from AC recommendations